# Aqueous Partition of Methanolic Extract of *Dicranopteris linearis* Leaves Protects against Liver Damage Induced by Paracetamol

**DOI:** 10.3390/nu11122945

**Published:** 2019-12-04

**Authors:** Zainul Amiruddin Zakaria, Farah Hidayah Kamisan, Nurliana Mohd. Nasir, Lay Kek Teh, Mohd. Zaki Salleh

**Affiliations:** 1Department of Biomedical Sciences, Faculty of Medicine and Health Sciences, Universiti Putra Malaysia, Serdang 43400, Selangor, Malaysia; lavandula88@yahoo.com (F.H.K.); liananasir89@gmail.com (N.M.N.); 2Integrative Pharmacogenomics Institute (I-PROMISE), Faculty of Pharmacy, Universiti Teknologi MARA, Puncak Alam 42300, Selangor, Malaysia; tehlaykek2016@gmail.com (L.K.T.);

**Keywords:** *Dicranopteris linearis*, hepatoprotective, free radical scavenging activity, endogenous antioxidant enzymes, saponins, triterpenes, phenolic derivatives

## Abstract

This study aimed to determine the antioxidant and hepatoprotective activities of semi-purified aqueous partition obtained from the methanol extract of *Dicranopteris linearis* (AQDL) leaves against paracetamol (PCM)-induced liver intoxication in rats. The test solutions, AQDL (50, 250, and 500 mg/kg), were administered orally to rats (*n* = 6) once daily for seven consecutive days followed by the hepatotoxicity induction using 3 g/kg PCM (p.o.). Blood was collected for serum biochemical parameters analysis while the liver was collected for histopathological examination and endogenous antioxidant enzymes analysis. AQDL was also subjected to antioxidant determination and phytochemical analysis. Results obtained show that AQDL possessed high total phenolic content (TPC) value and remarkable radical scavenging activities. AQDL also significantly (*p* < 0.05) reduced the liver weight/body weight (LW/BW) ratio or serum level of ALT, AST, and total bilirubin while significantly (*p* < 0.05) increase the level of superoxide dismutase (SOD) and catalase (CAT) without affecting the malondialdehyde (MDA) in the liver indicating its hepatoprotective effect. Phytoconstituents analyses showed only the presence of saponins and triterpenes, but lack of flavonoids. In conclusion, AQDL exerts hepatoprotective activity via its high antioxidant potential and ability to modulate the endogenous enzymatic antioxidant defense system possibly via the synergistic action of saponins and triterpenes.

## 1. Introduction

Drug-induced liver injury (DILI) is an infrequent but potentially life-threatening, undesirable drug reaction. It is the most common reason for drug withdrawal from the pharmaceutical market due to its association with significant adverse effects, morbidity, and mortality. A number of drugs, including bromfenac and troglitazone, have been removed from the market because of DILI [1]. DILI is responsible for the majority of acute liver failure cases and is now the leading cause for liver transplantation among patients. For example, DILI is accountable for about 10% of all adverse drug reactions in the United States (USA). Estimates of the annual incidence of DILI are reported to be as high as 14–24 cases/100,000 individuals and approximations show that almost 44,000 patients per year will develop DILI [2]. 

One of the important drugs related to DILI is paracetamol (PCM), a drug commonly utilized worldwide for its antipyretic or analgesic properties [3]. PCM is easily accessible in various formulations as an over-the-counter medication reflected by the fact that it is regularly consumed by over 60 million Americans on a weekly basis, making it the most widely utilized analgesic and antipyretic in the USA [4]. PCM has been reported to be one of the most common pharmaceutical products which cause DILI. Hepatic injury and subsequent hepatic failure due to both intentional and non-intentional overdose of PCM has affected patients for decades and remains a global issue [3]. PCM accounts for more than 50% of overdose-related acute liver failure and approximately 20% of the liver transplant cases [5]. Moreover, around 30,000 patients are admitted to hospitals every year for treatment of PCM hepatotoxicity in the USA. Mortality rates have been approximated at 0.4% in overdose patients, translating to 300 deaths annually in the USA [3].

According to the United States Food and Drug Administration (USFDA), PCM is considered non-toxic to an adult if consumed up to 1000 mg every 4−6 h, not to exceed 4000 mg day−1, for no longer than 10 days [6]. At therapeutic levels, PCM is principally metabolized in the liver via the processes of glucuronidation and sulphation to unreactive metabolites with a small fraction undergoes bioactivation via a process mediated by cytochrome P450 (CYP450) to form reactive toxic electrophiles known as N-acetyl-p-benzoquinoneimine (NAPQ1). NAPQ1 is then rapidly deactivated by an intracellular natural antioxidant known as glutathione (GSH). However, an overdose of PCM will result in the rise of toxic NAPQI metabolite generation, which extensively depletes the hepatocellular GSH level [7]. Excessive NAPQ1 production will lead to oxidative stress, which may exceed the antioxidant defense system, particularly GSH, resulting in the reduction of protective physiological moieties and decrease in repair capacity resulting in increased susceptibility of the liver to oxidative stress. With regard to the mechanism of action, these toxic metabolites will covalently bind to macromolecules in the vital biomembranes ensuing extensive propagation of the alkylation as well as peroxidation processes that alter cellular proteins, provoke oxidative stress, and cause damages to mitochondria leading to hepatocyte death. Other than that, NAPQ1 also induces lipid oxidation and alters the homeostasis of calcium. Membrane lipid peroxidation is directly related to the depletion of tissue GSH (an intracellular antioxidant) leading to the altered functional integrity of these structures, and if the damage is severe, it could be fatal [8]. Membrane lipid peroxidation may lead to alteration in membrane fluidity and permeability, enhanced rates of protein degradation, and ultimately hepatocytes death and liver injury. The concentration of intracellular GSH, therefore, is the key determinant of membrane integrity and the extent of toxicant-induced hepatic cell injury. The vital roles of reactive oxygen species (ROS) in the cellular damage are widely investigated and it has been suggested that the covalent binding of ROS, as well as reactive intermediates to macromolecules, could likely contribute to the severe harmful drug reactions. There are several studies that suggest the generation of reactive metabolites and free radicals from hepatotoxic drugs. Based on the above explanation, several main causes of the hepatotoxic reactions by drugs are elevated ROS generation, oxidative stress, and suppressed immune responses [9]. 

Currently, management of DILI, particularly of PCM-induced hepatotoxicity, is still a challenge to the modern medicine and controversial because the available pharmacological interventions, either conventional or synthetic drugs, are inadequate and occasionally cause serious side effects, which include being carcinogenic in long term use [10,11]. The increasing numbers of patients with liver dysfunction due to overwhelming usage of alcohol and drugs has paved the path for researchers to search for potential sources of new therapeutic agents for the prevention of DILI [12,13]. These plants are rich in triterpenes, flavonoids, or polyphenols, which have been now established as powerful hepatoprotective and chemopreventive agents in experimental liver-injury cell and animal models. Furthermore, as oxidative stress has been proven to play a major role in DILI, compounds with antioxidant activity might plausibly be good agents to reduce DILI [9]. The dietary nature, easy availability and less adverse reactions of medicinal plants provide them an extra edge over other candidates as supplements [14]. The basis behind the protection provided by the medicinal plants is hypothesized to be through their ability to remove free radicals from the cellular environment and therefore provide protection against ROS mediated damage to membrane lipids and macromolecules. Other than that, the protective potential of medicinal plants could also be attributed to their phytoconstituents ability to enhance endogenous antioxidants such as GSH and SOD biosynthesis/bioactivity, as well as to interact with various cytochrome P450 isoforms and to inhibit the entry of toxins to the cells [11]. It is envisaged that plant-based natural products will not only lower the risk of DILI but also provide an alternative solution to cure DILI.

One of the plants proven to possess medicinal values of remarkable antioxidant and anti-inflammatory activities is *Dicranopteris linearis* L. (family Gleicheniaceae) [15,16,17]. Known as “Resam” to the Malay, *D. linearis* is a fern widely distributed in Malaysia. Although not many traditional uses of this plant were recorded in Malay folklore medicine, its leaves were used as tonic or poultice to reduce body temperature [15,16]. Scientifically, *D. linearis* extracts have been reported to exert antinociceptive [15,16,18], antipyretic [16], anticancer [19,20], cytotoxic [17], hepatoprotective [21,22,23,24,25], and chemopreventive [26] activities. With regard to the hepatoprotective activity of *D. linearis*, preliminary studies using the chloroform, aqueous and methanol extracts of *D. linearis* leaf have successfully proven these extracts ability to attenuate PCM- and carbon tetrachloride-induced hepatotoxicity [21,22,24]. Further analyses by Kamisan et al. [23] revealed that the methanol extract of *D. linearis* leaf (MEDL) possessed remarkable antioxidant activity when assessed using the 2, 2-diphenyl-1-picrylhydrazyl (DPPH)- and superoxide anion-radical scavenging assays, oxygen radical absorbance capacity (ORAC) test and contained high total phenolic content (TPC) value, while Zakaria et al. [25] successfully demonstrated the ability of MEDL to enhance the endogenous antioxidant activity in PCM-induced hepatotoxic rats. 

Although several known bioactive compounds have been identified from MEDL [23,25] using the high pressure liquid chromatography (HPLC) and ultra-high pressure liquid chromatography-electrospray ionization coupled with high resolution mass spectrometry (UHPLC-ESI/HRMS) approaches, it is not possible to directly foretell the character of bioactive compounds that really contribute to the hepatoprotective effect of *D. linearis*. This could be due to the fact that methanol, being the solvent system for soaking *D. linearis* leaves, has the ability to dissolve all types of bioactive compounds regardless of their polarity. Therefore, attempt was made to partition the MEDL using several solvents of different polarity to obtain the petroleum ether, ethyl acetate, and aqueous partitions, which represent non-polar, intermediate polar and highly polar types of bioactive compounds, respectively. Since there is a need to ascertain the effectiveness of each partition to attenuate liver injury, all partitions were subjected to the preliminary screening against the PCM-induced hepatotoxic effect in rats. Based on the preliminary findings (data not shown), the present study was designed to explore the hepatoprotective potential of the aqueous partition (AEDL) using the same animal models as reported by Zakaria et al. [25].

Taking these findings into consideration, the present study was designed to partition the MEDL sequentially with petroleum ether, ethyl acetate, and distilled water, and the aqueous partition of MEDL (AQDL) was subjected to hepatoprotective study against the PCM-induced liver injury in rats.

## 2. Materials and Methods

### 2.1. Plant Material and Preparation of Methanol Extract *D. Linearis* (MEDL)

The wildly grown leaves of *D. linearis* were collected around the area of Serdang, Selangor, Malaysia, between February and March 2012. The leaves have been previously identified and a voucher specimen (SK 1987/11) has been deposited at the Herbarium of the Institute of Bioscience, Universiti Putra Malaysia (UPM). 

The leaves were washed with water to remove dirt and then dried in an oven at 40 °C for two weeks. During the drying time, the leaves were periodically turned over to provide uniform drying. The dried leaves obtained were ground to a coarse powder using a mill machine (CGOLDENWALL, China). Methanol extract of *D. linearis* (MEDL) was prepared as previously described in detail by Kamisan et al. [23]. Briefly, 160 g of powdered leaves were soaked in absolute methanol (1:20 (w/v)) for 72 h at room temperature and this procedure was repeated for three times. 

### 2.2. Preparation of Aqueous Partitions of MEDL (AQDL)

The preparation of semi-purified aqueous partition was carried out according to the similar procedure described elsewhere [27]. About 20 g of the dried MEDL was suspended in 1000 mL of methanol (MeOH; Fisher Scientific, UK), and then 200 mL of distilled water was added. The aqueous MeOH extract was transferred into a 2000 mL separatory funnel and 700 mL of petroleum ether (PE; Fisher Scientific, UK) was then added. The mixture was vehemently shaken and then left for 24 h to allow the mixture to separate into two-phase immiscible liquid solutions. The lower phase supernatant, which represents PE partition, was collected while the upper layer supernatant was further partitioned using new PE for another two times. All collected PE supernatants were pooled together, filtered using the Whatman No. 1 filter paper and then concentrated using the rotary evaporator (Buchi Rotavapor® R210/215) (40 °C; under reduced pressure (204 mbar)) to obtain the PE partition of MEDL (PEDL). Then, the residue of mixture solution was partitioned with ethyl acetate (EA; Fisher Scientific, UK) using the same partitioning procedure as described for PE to obtain the EA partition of *D. linearis* (EADL). The remaining upper layer residue was then subjected to further partitioning using ethyl acetate as described for PE partition and evaporated to obtain the EA partition of MEDL (EADL). Lastly, the remaining residual supernatant of aqueous MEOH solution was subjected to freeze-drying process to obtain the aqueous partition of MEDL (AQDL). AEDL was kept at −4 °C until further use while the other partitions were kept at −80 °C for future use.

### 2.3. Determination of the Antioxidant Activity of AQDL

#### 2.3.1. Total Phenolic Content 

Total phenolic content (TPC) of AQDL was verified according to the slightly modified procedure of Singleton and Rossi [28] as described in detail by Kamisan et al. [23]. Briefly, AEDL, at the concentration of 1 mg/mL, was mixed at room temperature with 80% methanol containing 1% hydrochloric acid and 1% distilled water and placed on a shaker set at 200 rpm for 2 h. The mixture was then centrifuged at 2817× *g* for 15 min to obtain the supernatant, which was mixed at the volume of 200 μL with 400 μL (0.1 mL/0.9 mL) of Folin–Ciocalteu reagent and permitted to stand for 5 min at room temperature. Subsequently, 400 μL of sodium bicarbonate (60 mg/mL) solution was added and the mixture was left to stand for 90 min at room temperature before the absorbance was read at 725 nm using a spectrophotometer. The level of TPC in the samples, expressed in terms of gallic acid equivalent (GAE) (mg GAE/g) dry extract, was measured based on the calibration curve created using the gallic acid standard optical density. The concentration of phenolics (mg/mL) was determined from the calibration line based on the measured absorbance.

#### 2.3.2. 2-Diphenyl-1-Picrylhydrazyl (DPPH) Radical Scavenging Assay

The free-radical scavenging potential of AQDL was determined using the modified method of Blois [29] as described by Kamisan et al. (2014). The partition in the concentration ranging from 3.13 µg/mL to 200 µg/mL was prepared from the stock solution (1 mg/mL). Briefly, a mixture of 50 µL of AQDL, 50 μL DPPH (1 mM in ethanolic solution) and 150 μL absolute ethanol (AR grade) was added in a 96-well microtiter plate and shaken for 15 s at 500 rpm. The plate was then kept at room temperature for 30 min followed by the measurement of absorbance at 520 nm. These procedures were carried out in triplicate. Based on the absorbance recorded, the concentration of DPPH radical was measured based on the given equation: DPPH scavenging effect (%) = [(A_O_ − A_X_)/(A_O_ − A_I_)] × 100,
where A_O_ is the absorbance of negative control, A_I_ is the absorbance of positive control, and A_X_ is the absorbance of the sample. 

The IC_50_ value was calculated from the linear part of the graph of the inhibition of DPPH radical [30]. Data analysis was done by using Graph pad PRISM V5.01. 

#### 2.3.3. Superoxide Anion Radical Scavenging Assay

The superoxide anion radicals scavenging action of AQDL was determined based on the modified procedure of Liu et al. [31] as described in detail by Kamisan et al. [23]. According to this modified procedure, the superoxide radicals were generated in phenazine methosulphate-nicotinamide adenine dinucleotide (PMS-NADH) systems through the oxidation of NADH and measured based on the reduction of nitroblue tetrazolium (NBT). Briefly, the reaction mixture containing the superoxide radicals was prepared by mixing 3 mL of Tris-HCl buffer (16 mM, pH 8) containing 1 mL of NBT (50 μM), 1 mL NADH (78 μM) and AQDL (25–50 μg) with 1 mL of phenazine methosulphate (PMS) solution (10 μM). The reaction mixture was then incubated at 25 °C for 5 min followed by the absorbance measurement at 560 nm using a spectrophotometer (UV–vis 1700, Shimadzu, Japan). Changes in the absorbance were compared between the AQDL-treated reaction mixtures against blank control or L-ascorbic acid as the positive control, and a decrease in the recorded absorbance indicates increasing superoxide anion scavenging activity. The percentage inhibition of superoxide anion production was evaluated using the following equation: Inhibition of superoxide anion generation (%) = [1 − (A_T_/A_C_)] × 100,
where A_C_ was the absorbance of the blank control and A_T_ was the absorbance in the presence of AQDL or positive control.

#### 2.3.4. Oxygen Radical Absorbance Capacity (ORAC) Test

The antioxidant capacity of AQDL was measured using the slightly modified oxygen-radical absorbance-capacity (ORAC) assay [32] as described in detail by Kamisan et al. [23]. Briefly, a peroxyl-radical generator was prepared daily by mixing 2,2-azobis (2-amidinopropane) dihydrochloride (AAPH) with 10 mL of 75 mM phosphate buffer (pH 7.4). On the other hand, 1 mM of sodium fluorescein stock solution was prepared by dissolving it in 75 mM phosphate buffer (pH 7.4) and stored in wrapping foil at 5 °C. Prior to its usage, the sodium fluorescein stock solution was further diluted 1:100,000 with 75 mM phosphate buffer (pH 7.4). Then, 150 μL of working solution of sodium fluorescein was added with 25 μL of Trolox or 25 μL of AQDL into the blank or sample wells of the 96-well microplate, respectively, and allowed to equilibrate by incubating for 10 min at 37 °C. After equilibration, the wells containing the respective mixture solution were added with 25 μL of 240 mM AAPH solution to initiate the reactions. The fluorescence intensity of each well was then determined kinetically with data taken every 1 min for 3 h using the BMG Omega Fluostar Fluorescence Spectrophotometer (BMG LABTECH, Ortenberg Germany), which was equipped with an excitation filter of 485 nm and an emission filter of 520 nm. ORAC values were calculated using the MARS Data Analysis Reduction Software.

### 2.4. Determination of the Anti-Inflammatory Activity of AQDL

#### 2.4.1. Lipoxygenase Assay

The effect of lipoxygenase (LOX) in the modulation of AQDL-induced hepatoprotective activity was determined using the slightly modified in vitro LOX assay [33] as described in detail by Kamisan et al. [23]. Briefly, a mixture solution containing 160 mL of sodium phosphate buffer (0.1 M, pH 8.0), 10 mL of AQDL (dissolved in MeOH), and 20 mL of soybean LOX solution was incubated for 10 min at 25 °C. Then, 10 mL of sodium linoleic acid solution, which act as the substrate, was added to instigate the reaction wherein linoleic acid was enzymatically converted to (9Z,11E)-(13S)-13-hydroperoxyoctadeca-9,11-dienoate. This reaction leads to the change in absorbance, which was measured using the spectrophotometer at 234 nm over the period of 6 min. All reactions were completed in triplicates in a 96-well microplate.

#### 2.4.2. Xanthine Oxidase Assay

The effect of xanthine oxidase (XO) in the modulation of AQDL-induced hepatoprotective activity was determined using the slightly modified in vitro XO assay [34] as described in detail by Kamisan et al. [23]. Briefly, a mixture solution containing 10 μL of AQDL (dissolved in DMSO), 10 μL of XO solution, and 130 μL of potassium phosphate buffer (0.05 M, pH 7.5) was prepared and incubated for 10 min at 25 °C. Then, 100 μL of xanthine solution, which act as the substrate, was added to start the reaction wherein xanthine was enzymatically converted to uric acid and hydrogen peroxides. This reaction led to the change in absorbance, which was measured using the spectrophotometer at 295 nm. All reactions were completed in triplicates in a 96-well microplate.

### 2.5. Experimental Animals

Male Sprague Dawley rats weighing between 180 and 200 g and 8–10 weeks old were obtained from the Animal Source Unit, Faculty of Veterinary Medicine (FVM), Universiti Putra Malaysia (UPM) and housed in the Animal House Unit, Faculty of Medicine and Health Sciences, UPM under the standardized environmental conditions for 48 h as described in detail by Kamisan et al. [23]. The study protocol of the present study has been approved by the Animal House and Use Committee, Faculty of Medicine and Health Sciences, UPM (Ethical approval no.: UPM/FPSK/PADS/BR-UUH/00449) as previously described [23].

### 2.6. Hepatoprotective Assay

The animals were randomly divided into six equal groups (*n* = 6) and fasted overnight prior to the experiment. Group 1 (Normal control) was pretreated (p.o) with vehicle (10% of dimethyl sulfoxide (DMSO)); Group 2 (Hepatotoxic control) was also pretreated (p.o) with vehicle; Group 3 (Positive control) was pretreated (p.o.) with 200 mg/kg silymarin suspended in 1% carboxymethyl cellulose (CMC), and; Groups 4–6 (Test groups) were pretreated (p.o.) with 50, 250, and 500 mg/kg AQDL suspended in 10% DMSO, respectively. Each group of the rats received the respective dose of test solution orally once daily for 7 consecutive days and 3 h after the last administration of test solution on day 7; Group 1 was treated (p.o.) with 10% DMSO whereas Groups 2–6 were treated (p.o.) with 3 mg/kg PCM. The rats’ body weight was taken prior to and after the administration of test solutions, and once daily for the next 7 days prior to the PCM treatment. Forty-eight h after the administration of PCM, rats from each of the groups were anesthetized by ketamine (100 mg/kg; intramuscular (i.m.)) and xylazine (16 mg/kg; i.m.), and blood was collected into heparinized bottles by cardiac puncture. Then the rats were euthanized by cervical dislocation and the liver was instantly removed, washed in ice-cold saline to remove the blood, and duly weighed. A segment from the midpoint lobe of the liver was conserved in 10% formalin solution for histological analysis. The remaining of the liver was promptly frozen in dry ice and stored at −80 °C for further endogenous antioxidant enzymes analysis. The experimental grouping of the rats is shown in Table 1.

#### 2.6.1. Biochemical Analysis of Collected Blood Samples

For the biochemical analyses, the blood samples were separated by centrifuging at 3000 rpm for 10 min, and the plasma samples were aspirated off and frozen at −80 °C until use [23]. Later, the plasma samples were biochemically analyzed to determine the level of alanine aminotransferase (ALT), aspartate aminotransferase (AST), alkaline phosphatase (ALP), and total bilirubin (TB) using the Hitachi 902 Automatic Chemical Analyser (Hitachi, Minato-ku, Tokyo, Japan). 

#### 2.6.2. Determination of the Activity of Antioxidant Enzymes (Superoxide Dismutase (SOD) and Catalase (CAT)) and Level of Malondialdehyde (MDA) in Liver Homogenates

Liver tissues were homogenized in cold phosphate buffer (5 mM, pH 7.4) to give a 10% (w/v) homogenate and then centrifuged at 4000 rpm for 25 min at 4 °C. Prior to the measurement of SOD, CAT, and MDA activities, protein concentration was determined using the Bradford method [35] with bovine serum albumin (BSA) used as the standard. Later, the supernatant of liver homogenate was subjected to the estimation of SOD and CAT activity and MDA level using the respective manufacturer’s protocols provided within the commercial kits (Cayman Chemical Company, Ann Arbor, MI, USA).

#### 2.6.3. Histological Analysis of the Treated Liver

The livers were routinely processed in an Automatic Tissue Processor (Leica TP1020, Nussloch, Germany), and then embedded in paraffin wax with Leica EG 1160 (Leica Microsystems, Nussloch, Germany). Then, the tissues were sectioned to a 5–6 mm thickness using a microtome (Leica RM2125 RTS, Nussloch, Germany) and then stained with hematoxylin and eosin dye using the Tissue- Tek Prisma- Ezs Autostainer (Sakura, Torrance, CA, USA). The stained section was then subjected to the microscopic examination using the light microscope (Olympus-CX31, Shinjuku, Tokyo, Japan). The liver sections were then scored and evaluated by a pathologist according to the severity of the hepatic injury as described by El-Beshbishy et al. [36].

### 2.7. Analyses of Phytochemical Constituents of AQDL

#### 2.7.1. Phytochemical Screening of AQDL

AQDL was subjected to the qualitative phytochemical screening for the presence of saponins, steroids, flavonoids, triterpenes, tannins, and alkaloids according to procedures described by Ikhiri et al. [37]. 

#### 2.7.2. High-Performance Liquid Chromatography (HPLC) Analysis of AQDL

The partition (AQDL) was subjected to HPLC analysis using the following HPLC system; Waters Delta 600 with 600 Controller equipped with Waters 996 photodiode array detector and a Phenomenex Luna column (5 µm; 4.6 mm i.d.  × 250 mm) (Torrance, CA, USA). The procedures used were as reported by Zakaria et al. [38]. In brief, two types of eluants namely 0.1% aqueous formic acid (eluant A) and acetonitrile (eluant B) were used. The gradient elution was programmed so that the initial condition was 95% A and 5% B (acetonitrile) with a linear gradient reaching 25% B at *t* = 12 min. After 10 min (*t* = 22 min), the gradient of B was decreased to 15% and sustained for 12 min (*t* = 30 min). Starting from *t* = 30 min, the program was returned to the initial solvent composition at *t* = 35 min. Millennium 32 Chromatography Software (Waters Co., Milford, MA, USA) was used to record and integrate the chromatograms at peak areas with the peak elution monitored at various wavelengths, namely 210, 254, 280, 300, 330, and 366 nm. The verified chromatograms were analyzed, and the recorded retention times, peak areas, and UV spectra of main peaks were studied. The HPLC profiles of AQDL were also analyzed by comparing the chromatogram of AQDL against the respective chromatogram of several pure bioactive compounds available in the laboratory, namely fisetin, quercetin rutin, quercitrin, naringenin, genistein, pinostrobin, hesperetin, flavanone, 4′,5,7-trihydroxyflavanone, 2,4,4′-trihydroxychalcone, dihydro quercetin or hesperetin at 254 nm. Retention times for each of the peak of standard pure compounds were compared against the peaks of AQDL, and the similarity between these retention times and UV spectra information recorded indicated the presence of the particular bioactive compounds.

### 2.8. Statistical Analysis

Data were expressed as mean values ± SD of six rats in each group. Statistical analysis was performed using one-way analysis of variance (ANOVA) followed by Dunnet’s Multiple Comparison test with *p* < 0.05 considered to be statistically significant. Statistical analysis was conducted using the Graph Pad Prism software version 5. 

## 3. Results 

### 3.1. Extraction Yield of AQDL

Approximately 74.0 g of crude MEDL was partitioned using solvents of increasing polarity to obtain semi-purified petroleum ether (3.38 g), ethyl acetate (8.59 g), and aqueous (11.27 g) partitions of *D. linearis*. The aqueous partition of *D. linearis* (AQDL) was chosen to be further tested for its antioxidant, anti-inflammatory, and hepatoprotective potential. 

### 3.2. TPC Value, Free Radical Scavenging-, and Antioxidant-Activity of AQDL

The TPC value and radical scavenging activities of AQDL are showed in Table 2. AQDL was also found to possess characteristics of a strong antioxidant agent indicated by the high TPC value (>100 mg GAEs/100 mg extract), remarkable DPPH- and superoxide-radical scavenging activities (75% to 85% scavenging activity in comparison to the standard drug, 200 µg/mL ascorbic acid or 200 µg/mL SOD, respectively), and high ORAC value (>15,000 µmol TE/ 100 g in comparison to the standard drug, 200 µg/mL Trolox). 

### 3.3. Effect of AQDL against the LOX- and XO-Mediated Inflammatory Activity 

Table 3 shows the in vitro effect of AQDL against the in vitro LOX- and XO-mediated inflammatory activity. At 100 µg/mL, AQDL induced a very low inhibitory effect on the LOX-mediated inflammatory activity (<20% inhibition) while failed to inhibit the XO-mediated inflammatory activity. 

### 3.4. In Vivo Hepatoprotective Activity of AQDL

The effect of AQDL on the body weight (BW), liver weight (LW), and LW/BW ratio; the level of serum biochemical parameters (serum ALT, AST, ALP, and TB); and the activity of endogenous antioxidant enzymes (SOD and CAT) or lipid peroxidation marker (MDA) in PCM-induced hepatotoxic rats are shown in Table 4, Table 5 and Table 6, respectively. In addition, the histopathological examination of the PCM-intoxicated liver tissues pretreated or non-pretreated with AQDL is shown in Figure 1. 

#### 3.4.1. Effect of AQDL on the Body Weight (BW), Liver Weight (LW), and the LW/BW Ratio of PCM-Intoxicated Rats

From the results obtained, although there were no significant changes in their BW when compared to the normal group, the PCM-induced hepatotoxic rats demonstrated significant (*p* < 0.05) increases in their LW and LW/BW ratio, which was significantly (*p* < 0.05) reduced when pre-treated with all doses of AQDL or silymarin (Table 4). 

#### 3.4.2. Effect of AQDL on the Serum Level of ALT, AST, ALP, and TB of PCM-Intoxicated Rats

Moreover, the levels of serum ALT, AST, ALP, and total bilirubin were found to be abnormally high in PCM-intoxicated group indicating liver dysfunction and signified the injury to hepatocytes (Table 5). Interestingly, AQDL significantly (*p* < 0.05) reversed the increased in serum enzymes level induced by PCM suggesting its ability to hinder leakage of intracellular enzymes through its membrane stabilizing activity. In addition, the significant (*p* < 0.05) reduction in serum TB level following AQDL pre-treatment in PCM-intoxicated rats indicated the effectiveness of AQDL in returning the liver to its normal functional status (Table 5). 

#### 3.4.3. Effect of AQDL on the Activity of Endogenous Enzymatic Antioxidant System, Namely, SOD and CAT, of PCM-Intoxicated Rats

Incidentally, PCM intoxication caused significant (*p* < 0.05) decrease in SOD and CAT activities, and these effects were significantly (*p* < 0.05) reversed following the AQDL or silymarin pre-treatment (Table 6). However, the present study also shows that AQDL did not significantly (*p* < 0.05) affect the serum level of MDA, which was significantly (*p* < 0.05) increased following PCM intoxication. 

### 3.5. Histopathological Findings on the Effect of AQDL against the PCM-Intoxicated Liver Tissue Section 

Figure 1 shows the histopathological findings of PCM-intoxicated liver tissues with pre-treatment or without pretreatment with AQDL. PCM-intoxicated liver tissues (Figure 1b) demonstrated the most severe damage in hepatocyte architecture such as massive hemorrhagic necrosis around the centrilobular region, severe cytoplasmic vacuolation, broad infiltration of lymphocytes around the central vein and in the portal areas, loss of cellular boundaries, and ballooning degeneration in comparison to the liver tissue of normal control group (Figure 1a), which demonstrated the normal cellular architecture indicated by distinct hepatic cells that were radiantly arranged around the central vein, the hepatocytes with prominent nucleus and well-preserved cytoplasm, and the well-defined sinusoidal line. Interestingly, pre-treatment with silymarin (Figure 1c) or AQDL (Figure 1d–f) attenuated PCM toxic effect as indicated by the presence of a relatively normal lobular pattern with a mild degree of necrosis and lymphocyte infiltration observed. Table 7 shows the histopathological scoring of the liver section of PCM-induced hepatotoxic rats with or without pretreatment with AQDL. Overall, the histological findings corroborate well with the serum biochemical and endogenous antioxidant enzymes activities and further confirmed that AQDL has the ability to reduce the degree of PCM-induced liver injury.

### 3.6. Phytochemical Analyses of AQDL

The qualitative phytochemical screening of AQDL revealed only the presence of saponins and triterpenes (Table 8). In addition, the HPLC analysis of AQDL at 210, 254, 280, 300, 330, and 366 nm revealed the phytoconstituents profiles of AQDL (Figure 2A). Two different peaks were detected at 366 nm, namely, (i) peak 1, detected at 254–366 nm with the recorded retention time (RT) of approximately 19.311 min, exerted the absorption peak (UV–vis spectral) of 264.9 nm for Band I and 347.0 nm for Band 2; and (ii) peak 2, detected at 254–366 nm with the recorded RT of approximately 19.925 min, exhibited the UV–vis spectral at 227.2 nm (Band I) and 313.6 nm (band 2) (Figure 2A). 

Furthermore, comparison of the detected peaks of AQDL at 366 nm with the peak of several pure flavonoid-based compounds showed that none of the peak of the standard flavonoids matched any of the peaks of AQDL. Figure 2B showed only a comparison between the chromatogram of AQDL [Peak 1 (RT = 19.311 min) or Peak 2 (RT = 19.925 min)] against the chromatogram of rutin (RT = 20.395 min) and quercetin (RT = 27.472 min). 

## 4. Discussion

Phenolics, the most extensive secondary metabolites in the plant kingdom, have received much interest as prospective natural antioxidants in terms of their abilities to act as both efficient radical scavengers and metal chelators [39]. The high TPC value of AQDL corresponded well with our previous finding [17] and further supported the report made by Lim and Lai [40] on the high TPC content of *D. linearis*, thus, could help to explain the remarkable radical scavenging activities demonstrated by AQDL. Although flavonoids and tannins were not detected during phytochemicals screening of AQDL, the high TPC value could be attributed to the presence of small number of flavonoids as detected during the HPLC analysis and/or possibly by the presence of derivatives of phenolic compounds such as phenolic acid esters and phenolic *esters of* triterpenoids [41,42,43]. Other than flavonoids or phenolic-based compounds, saponins themselves have also been reported to possess radical scavenging activity and antioxidant activity [44,45,46]. Based on the established reports associating oxidative stress as part of the mechanisms of liver injury [47], it is reasonable to suggest that *t*he remarkable radical scavenging and antioxidant activities of AQDL contribute to its significant hepatoprotective activity. These activities of AQDL thwart the redox state precipitated intracellularly due to the action of NAPQ1 and hence ensure hepatoprotection against PCM-induced liver injury. The findings of this study corroborate the effect that was reported for MEDL [23,25].

Despite the good antioxidant activity, AQDL exerts low inhibitory effect in both in vitro LOX and XO inflammatory assay. The LOXs pathway is involved in the metabolism of leukotrienes and has been implicated in hepatic inflammation and liver damage [48]. Interestingly, further studies by Li et al. [49] revealed that inhibition of LOX pathway attenuates acute liver failure by inhibiting macrophage activation, while Pu et al. [50] demonstrated the critical role of 5-LO activity in PCM-induced liver injury by regulating paracetamol metabolism and oxidative stress. It was found that pharmacological inhibition of 5-LO in mice markedly ameliorated PCM-induced liver injury while inhibition of 5-LO induced hepatoprotective effect, which was associated with induction of the antitoxic phase II conjugating enzyme (sulfotransferase2a1), suppression of the pro-toxic phase I (CYP3A11), and reduction of the hepatic transporter (MRP3). The low inhibitory effect of AQDL against LOX activity suggested that AQDL did not exert hepatoprotective activity against PCM-induced intoxication via the inhibition of LOX-mediated inflammatory activity. This finding further corroborates the report made by Zakaria et al. [25] that MEDL also showed a low inhibitory activity against LOX-mediated inflammatory activity. Meanwhile, xanthine oxidase (XO) is an enzyme that generates ROS such as superoxide radicals and hydrogen peroxide when it catalyzes the oxidation of hypoxanthine to xanthine and can further catalyze the oxidation of xanthine to uric acid [51]. XO was further considered as a potential source of ROS in the liver after PCM overdose based partly on the ability of xanthine oxidase inhibitor, allopurinol, to attenuate the oxidative stress and liver injury in mice intoxicated with PCM [52]. However, further investigation by Williams et al. [53] has provided evidence that argues for a possible participation of xanthine oxidase, as a relevant source of ROS, in the pathophysiology of PCM-induced hepatotoxicity. However, according to Du et al. [54], it is obvious that the mechanism of liver protection is not simply the ability of allopurinol to inhibit xanthine oxidase or scavenge reactive oxygen but might involve alteration of the intracellular signaling pathways or up-regulation of the expression of cytoprotective genes. It was also suggested that mitochondria are the main source of ROS, which impairs mitochondrial function and is responsible for cell signaling resulting in cell death. Du et al. [54] also proposed that mitochondrial targeted antioxidants can be viable therapeutic agents against hepatotoxicity induced by PCM overdose, and re-purposing existing drugs to target oxidative stress and other concurrent signaling events can be a promising strategy to increase their potential application in patients with PCM overdose. Taking these findings and suggestions into consideration, it is plausible to suggest that AQDL did not modulate the XO-mediated pathway as indicated by its failure to produce inhibitory effect when assessed using the XO-mediated inflammatory assay. Thus, it is possible that the ability of AQDL to attenuate PCM-induced hepatotoxic effect could be attributed to its ability to scavenge ROS synthesized in mitochondria. However, this suggestion needs further in-depth study before a conclusion could be made. Although the present findings on the lack of anti-inflammatory activities of AQDL against LOX- and XO-mediated inflammation correspond well with the lack of effect of MEDL against LOX and XO actions [25], these observations seem to contradict the earlier reports on aqueous extract of *D. linearis*’ remarkable anti-inflammatory activity [19]. However, this discrepancy might be explained by the fact that the anti-inflammatory activity of aqueous extract of *D. linearis* was assessed using the in vivo cyclo-oxygenase-2 (COX-2)-mediated inflammatory model indicating the ability of the extract to inhibit COX-2-mediated inflammation [55]. Interestingly, the involvement of COX-2 in PCM-induced liver intoxication has been reported by Reilly et al. [56], who demonstrated that COX-2, but not COX-1, was induced in livers of PCM-intoxicated mice. It is commonly agreed that the mode of action of PCM is the inhibition of prostanoid biosynthesis, particularly prostaglandins (PGs), via the COX-mediated pathway of arachidonic acid metabolism. COX-1 and COX-2 are bifunctional enzymes that work via, firstly, COX activity and, secondly, peroxidase activity. It was through the peroxidase action that prostaglandins-G2 (PGG2) was converted to prostaglandin H2 (PGH2). On the other hand, PGH2 is converted by various, cell specific enzymes into final PGs, prostacyclin and thromboxane, collectively known as prostanoids. Prostanoids bind to and activate the prostanoid receptors, a G-protein coupled receptors (GPCRs), to obtain a vast range of responses including sensitization to pain, inflammation and fever, immune and cardiovascular functions. It is well acknowledged that NSAIDs and the specific COX inhibitors interfere with the COX activity whereas PCM interferes with the peroxidase activity. Coincidently, Suciu et al. [57] have reported on the activation of a COX-mediated signaling pathway, predominantly the COX-2-prostanoid pathway, during PCM-induced liver injury. Hence, the ability of AQDL to attenuate PCM-induced intoxication could plausibly be attributed in part to its inhibitory action on the COX-2-prostanoid pathway that involved inhibition of the peroxidase action.

Excess consumption of PCM either accidentally or non-accidentally can cause acute liver intoxication, which, if not properly treated, results in liver damage [1,2,3]. This acute toxic effect results from PCM being metabolized by the liver to form a toxic electrophile called NAPQ1 that binds covalently to tissue macromolecules, possibly oxidizes lipids or the essential sulfhydryl groups (protein thiols) and changes the homeostasis of calcium [58]. Excessive production of reactive species can contribute to diminution of defensive physiological moieties, such as GSH and α-tocopherol, resulting in damage to the macromolecules in imperative biomembranes and, finally, liver injury. Consequently, when there is liver injury, several enzymes, namely, ALT, AST, ALP, and molecules such as bilirubin and albumin, which are initially present in the cytoplasm, seep out into the blood stream [59]. Therefore, liver function can be evaluated by assessing the activities of serum ALT, AST, ALP, and bilirubin. In addition, according to Sharma et al. [60], the increase in serum level of ALP is due to its increase synthesis in the presence of increasing biliary pressure and reflects the pathological modification in biliary flow. In addition, verification of serum bilirubin also signifies an index for evaluation of hepatic function wherein any unusual increase in the levels of serum bilirubin point towards hepatobiliary-related ailments and severe disruption of hepatocellular function [61]. It is known that bilirubin is a metabolic product of hemoglobin and undergoes conjugation with glucuronic acid in hepatocytes to increase its solubility in water. Moreover, NAPQ1 actions also resulted in the decreased in the activity of endogenous enzymatic antioxidants such as SOD and CAT, which provide endogenous antioxidant defense by scavenging the superoxide anion to form hydrogen peroxide, therefore reducing the toxic effect caused by these radicals [8].

Enzymatic antioxidants such as SOD and CAT are involved in scavenging superoxide anion to form hydrogen peroxide, therefore reducing the toxic effect caused by these radicals. These enzymes are essential in the enzymatic antioxidant defense system, and declines in their activities may result in numerous toxic effects [62]. Natural antioxidants fortify the endogenous antioxidants defenses from ROS and reinstate the optimal balance by neutralizing reactive species [11]. The ability of AQDL to increase the hepatic SOD and CAT activities in PCM-intoxicated rats indicates that AQDL has the ability to diminish reactive free radicals, thereby reducing oxidative damage to the liver tissues by improving the activity of hepatic antioxidant enzymes.

Phytoconstituents of AQDL were investigated using the qualitative phytochemical screening and HPLC analysis. Based on the first assay, only saponins and triterpenes were detected in AQDL. This finding corroborates well the HPLC analysis, which revealed the presence of several small peaks with only one peak (Peak 1) suggested to be a flavonoid-based bioactive compound based on the UV–vis spectral recorded. Based on the characteristic UV–vis spectra of the two peaks detected at 366 nm, it is believed that only peak 1 represents a flavonoid-based compound based on the UV–vis spectral recorded (Band 1 = 347.0 nm; Band 2 = 264.9 nm) [63,64]. The chromatograms also clearly revealed that AQDL contains: (i) a low number of flavonoid-based bioactive compounds indicated by only one major peak detected with UV–vis spectral that fall in the range of 254–366 nm, and; (ii) a low content of flavonoid-based compounds as indicated by the presence of small size peak. These facts might explain why flavonoid was not detected during the qualitative phytochemical screening stage. It is believed that Peak 1 might possibly belong to the phenolic glycosides group particularly of the flavonol 3-O-glycosides group as reported by Raja et al. [65], Chen et al. [66], and Ponnusamy et al. [67]. Further phytoconstituents analyses need to be carried out and the possible bioactive compounds in AQDL need to identified. Overall, the phytochemical analyses revealed that the extract contained mainly saponins and triterpenes. Generally, the presence of phenolic-based compounds, saponins, and triterpenes were believed to contribute synergistically to the observed antioxidant and hepatoprotective activities of AQDL [68,69,70].

## 5. Conclusions

In conclusion, the present study suggests that AQDL possessed remarkable antioxidant activity and exerted hepatoprotective effect against PCM-induced intoxication partly via the activation of the endogenous antioxidant defense system and the presence of saponins, triterpenes and, possibly, phenolic derivatives. Overall, these findings could be used as a basis to further develop *D. linearis* as a hepatoprotective therapy for human usage.

## Figures and Tables

**Figure 1 nutrients-11-02945-f001:**
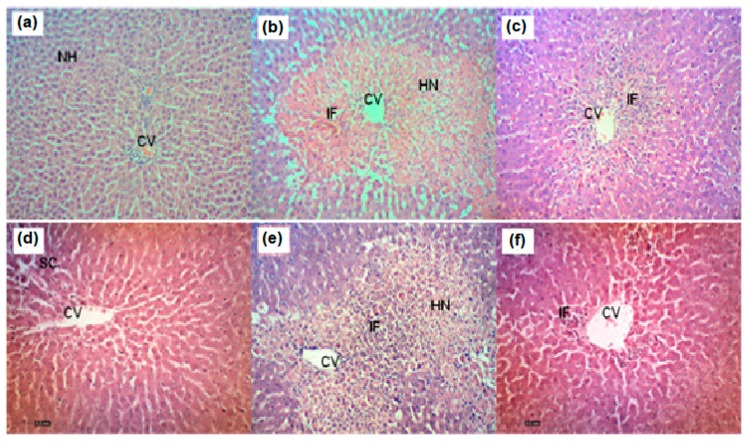
(**a**) Normal liver parenchyma. (**b**) Section of liver tissue treated with 3 g/kg PCM (p.o) showing large area of hemorrhagic necrosis around centrilobular region; inflammatory cell infiltration was observed at the center of the necrotic foci. (**c**) Section of liver tissue pre-treated with 200 mg/kg silymarin followed by PCM showing preservation of normal hepatocytes. (**d**) Section of liver tissue pre-treated with 50 mg/kg AQDL followed by PCM showing mild sinusoidal congestion and cellular swelling. (**e**) Section of liver tissue pre-treated with 250 mg/kg AQDL followed by PCM showing moderate hemorrhagic necrosis in centrilobular region and presence of inflammatory infiltrate. (**f**) Section of liver tissue pre-treated with 500 mg/kg AQDL followed by PCM showing mild inflammatory infiltrate and mild cellular swelling. (H&E staining, 100x magnification). CV = central vein. IF = inflammatory infiltrate. HN = hemorrhagic necrosis. SC = sinusoidal congestion. S = steatosis.

**Figure 2 nutrients-11-02945-f002:**
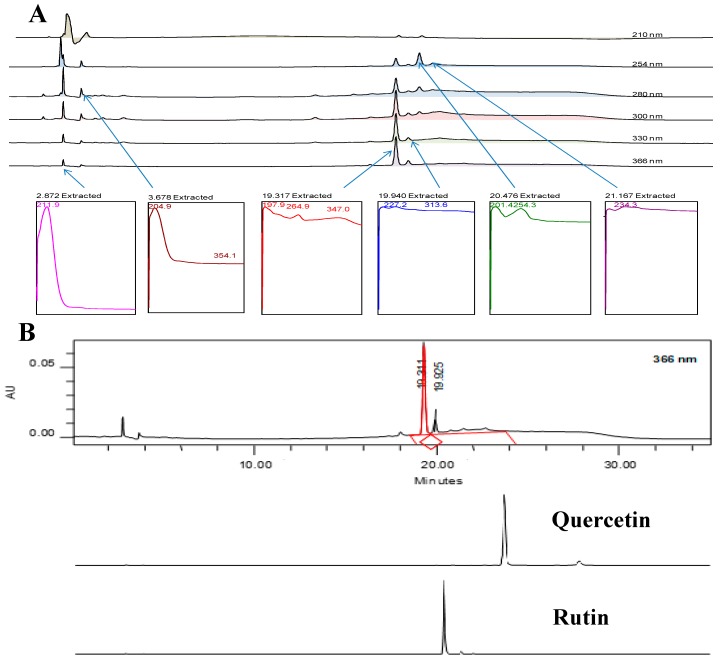
(**A**) HPLC profile of AQDL shows several detected peaks with their respective retention time and UV–vis spectral information at different wavelengths. Only two peaks were detected at 366 nm, and only one peak (RT = 19.327 min) possessed the UV–vis spectral (Band A fall in the range of 310–350 nm; Band B fall in the range of 250–290 nm) that is a characteristic of flavonoid-based (flavones type) bioactive compound. (**B**) Comparison between the chromatogram of AQDL against the chromatograms of several pure flavonoids revealed that none of the flavonoids were present in the said partition. Only two chromatograms of pure flavonoids, namely, quercetin and rutin, were included for comparison with AQDL.

**Table 1 nutrients-11-02945-t001:** Experimental grouping of the rats used to study the hepatoprotective activity of AQDL against PCM-induced hepatotoxic rats.

Group	Oral Pre-Treatment for 7 Days	3 Hours after Last Treatment on 7th Day
Normal control	10% DMSO	10% DMSO
Intoxicated group	10% DMSO	3 mg/kg PCM
Positive control	200 mg/kg silymarin	3 mg/kg PCM
Treatment	50 mg/kg AQDL	3 mg/kg PCM
	250 mg/kg AQDL	3 mg/kg PCM
	500 mg/kg AQDL	3 mg/kg PCM

**Table 2 nutrients-11-02945-t002:** TPC and free radical scavenging activity of 200 µg/mL AQDL.

Sample	Total Phenolic Content (TPC) ^1^	DPPH Radical Scavenging (%)	Superoxide Scavenging (%)	Total ORAC Value (µM TE/ 100 g)
Standard	Gallic acid (GAE)	Ascorbic acid (AA)	Superoxide dismutase (SOD) (6 × 10^−3^ U/mL)	Trolox standard curve
AQDL	193.5 ± 14.8	84.3 ± 2.6	79.0 ± 2.5	18997 ± 1096

Values are expressed in mean ± SEM. ^1^ Data expressed as TPC mg/100 g GAE are mean values of triplicate wells in duplicate experiments. Standard error of mean (SEM) < 5%.

**Table 3 nutrients-11-02945-t003:** In vitro anti-inflammatory effect of 100 mg/mL AQDL assessed using the lipoxygenase and xantine oxidase assays.

Sample	Lipoxygenase (%)	Xanthine Oxidase (%)
Sample concentration	100 mg/mL	100 mg/mL
AQDL	16.5 ± 1.3	NA

All values are expressed as mean ± SEM. Note: H, high (71% to 100%); M, moderate (41% to 70%); L, low (0% to 40%); NA, not active.

**Table 4 nutrients-11-02945-t004:** Effect of AQDL on the percentage of liver weight/body weight (LW/BW) ratio in PCM-treated rats.

Treatment	Dose (mg/kg)	Mean of Body Weight, BW (g)	Liver Weight, LW (g)	LW/BW (%)
Control	-	208.7 ± 5.6	5.9 ± 0.3	2.8 ± 0.1
DMSO + PCM		219.5 ± 4.7	9.7 ± 0.9 ^a^	4.4 ± 0.4 ^a^
Silymarin + PCM	200	200.0 ± 4.7	6.9 ± 0.2 ^b^	3.5 ± 0.1 ^b^
	50	166.2 ± 7.8	6.6 ± 0.4 ^b^	3.9 ± 0.1
AQDL + PCM	250	187.7 ± 2.2	8.0 ± 0.3 ^b^	4.2 ± 0.2
	500	189.8 ± 4.7	6.3 ± 0.3 ^b^	3.3 ± 0.1 ^b^

Values are expressed as means ± S.E.M. of six replicates. ^a^ Significant different as compared to normal control, *p* < 0.05. ^b^ Significant different as compared to negative control, *p* < 0.05.

**Table 5 nutrients-11-02945-t005:** Effect of AQDL on serum level of ALT (U/L), AST (U/L), ALP (U/L), and total bilirubin (µmol/L).

Treatment	Dose (mg/kg)	ALT (U/L)	AST (U/L)	ALP (U/L)	TB (umol/L)
Control	-	15.83 ± 2.9	95.13 ± 5.9	115.7 ± 7.0	0.5 ± 0.2
DMSO + PCM		1714 ± 142.2 ^a^	2266 ± 340.4 ^a^	330.0 ± 42.4 ^a^	4.1 ± 0.8 ^a^
Silymarin + PCM	200	474.5 ± 82.2 ^b^	690.9 ± 146.6 ^b^	195.5 ± 11.1 ^b^	2.3 ± 0.3 ^b^
	50	80.40 ± 10.3 ^b^	134.9 ± 22.4 ^b^	321.3 ± 4.3	0.9 ± 0.3 ^b^
AQDL+ PCM	250	908.4 ± 172.9 ^b^	1705 ± 403.5	329.7 ± 44.0	1.2 ± 0.5 ^b^
	500	298.1 ± 27.1 ^b^	527.6 ± 102.1 ^b^	286.8 ± 24.4	1.7 ± 0.3 ^b^

Values are expressed as means ± S.E.M. of six replicates. ^a^ Significant different as compared to normal control, *p* < 0.05. ^b^ Significant different as compared to negative control, *p* < 0.05.

**Table 6 nutrients-11-02945-t006:** Effects of AQDL on liver SOD, CAT, and MDA levels in PCM intoxicated rats.

Treatment	Dose (mg/kg)	SOD (U/g tissue)	CAT (U/g tissue)	MDA (µM)
Control	-	9.7 ± 0.4	114.8 ± 1.6	2.6 ± 0.6
DMSO + PCM		4.0 ± 0.1 ^a^	92.9 ± 1.9 ^a^	5.0 ± 0.6 ^a^
Silymarin + PCM	200	12.0 ± 0.2	109.5 ± 4.7 ^b^	2.6 ± 0.3 ^b^
	50	18.5 ± 0.2 ^b^	114.1 ± 0.8 ^b^	3.3 ± 0.6
AQDL + PCM	250	15.1 ± 0.4 ^b^	112.3 ± 1.2 ^b^	4.3 ± 0.6
	500	17.8 ± 0.1 ^b^	112.2 ± 1.8 ^b^	4.6 ± 0.6

Values are expressed as means ± S.E.M. of six replicates. ^a^ Significant different as compared to normal control, *p* < 0.05. ^b^ Significant different as compared to negative control, *p* < 0.05.

**Table 7 nutrients-11-02945-t007:** Effect of AQDL on the histopathological scoring of liver section of PCM-intoxicated rats.

Treatment	Dose (mg/kg)	Steatosis	Necrosis	Inflammation	Hemorrhage
Control	-	-	-	-	-
DMSO + PCM		+	+++	++	+
Silymarin + PCM	200	-	+	-	-
	50	-	+	-	-
AQDL + PCM	250	+	++	+	-
	500	+	+	-	-

The severity of various features of hepatic injury was evaluated based on those following scoring scheme: - normal, + mild effect, ++ moderate effect, +++ severe effect.

**Table 8 nutrients-11-02945-t008:** Qualitative phytochemical screening revealed the phytoconstituents of AQDL.

Sample	Phytochemical Constituents						Conclusion
	ALK	SAP	FLA	TAN	TTP	STR	
AQDL	-	+	-	-	+	-	Saponins and triterpenes only detected.

ALK: Alkaloids; SAP: Saponins; FLA: Flavonoids; TAN: Tannins; TTP: Triterpenes; STR: Steroids; +: detected; -: not detected.

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
