# Peer review of "Aqueous Partition of Methanolic Extract of Dicranopteris linearis Leaves Protects against Liver Damage Induced by Paracetamol"

_nutrients, 2019, doi:10.3390/nu11122945_

Round 1
Reviewer 1 Report
The manuscript: ‘Aqueous partition of methanolic extract of 2 Dicranopteris linearis leaves protects against liver 3 damage induced by paracetamol’ is a well written manuscript covering a very important issue. The study describe the antioxidant and hepatoprotective activities of semi-17 purified aqueous partition obtained from the methanol extract of Dicranopteris linearis (MEDL) leaves 18 against paracetamol (PCM) -induced liver intoxication in rats.
The manuscript could be published in its current form.The results are very important and valuable for developing interventions. I think the article is interesting for the readers of Nutrients. I just want to remark that it would be better to upload the tables within the margin limit.
Author Response
Journal: Nutrients
Manuscript ID: nutrients-600265
Manuscript title: Aqueous partition of methanolic extract of Dicranopteris linearis leaves protects against liver damage induced by paracetamol
Reviewer 1 Comments:
The manuscript: ‘Aqueous partition of methanolic extract of 2 Dicranopteris linearis leaves protects against liver 3 damage induced by paracetamol’ is a well written manuscript covering a very important issue. The study describe the antioxidant and hepatoprotective activities of semi-17 purified aqueous partition obtained from the methanol extract of Dicranopteris linearis (MEDL) leaves 18 against paracetamol (PCM) -induced liver intoxication in rats.
The manuscript could be published in its current form. The results are very important and valuable for developing interventions. I think the article is interesting for the readers of Nutrients. I just want to remark that it would be better to upload the tables within the margin limit.
RESPOND:
The authors have amended all tables to fit within the margin limit as requested by the reviewer.

Author Response
Journal: Nutrients
Manuscript ID: nutrients-600265
Manuscript title: Aqueous partition of methanolic extract of Dicranopteris linearis leaves protects against liver damage induced by paracetamol
Reviewer 2 Comments:
The study aimed to determine the antioxidant and hepatoprotective activities of semi-purified aqueous partition obtained from the methanol extract of Dicranopteris linearis (MEDL) leaves against paracetamol (PCM)-induced liver intoxication in rats. I found the paper to be overall well written and much of it to be well described. The design of the field campaign combined with several micro-met stations makes the dataset seem quite useful for the purpose. However, I found some obvious errors in the manuscript. I recommend that a minor revision is warranted. I ask that the authors specifically address each of my comments in their response.
Please provide information on cytotoxicity and IC50 of extract of Dicranopteris linearis leaves.RESPOND:
In the present study, no cytotoxicity testing was carried out on the extract of D. linearis. However, we have previously reported on the cytotoxic potential of aqueous, chloroform and methanol extracts of D. linearis leaves against the normal cell line (3T3) and several types of cancer cell lines (i.e. MCF-7, HL-60, HeLa, K562, HT-29, MD-MBA-231) (Zakaria et al., 2011). Interestingly, all extracts of D. linearis did not cause any cytotoxic effect on the normal 3T3 cell lines indicating that they were safe for consumption. Moreover, some of the extracts did show promising antiproliferative activity against some of the cancer cell lines with the recorded IC50 value that is lower than 30 ug/ml as requested by the UDFDA. Reference for this report is given below:
A. Zakaria, A. M. Mohamed, N. S. Mohd. Jamil, M. S. Rofiee, M. N. Somchit, A. Zuraini, A. K. Arifah and M.R. Sulaiman (2011). In vitro cytotoxic and antioxidant properties of the aqueous, chloroform and methanol extracts of Dicranopteris linearis leaves. African Journal of Biotechnology Vol. 10 (2), pp. 273-282.
In Figure 1, please explain why portions of figure description were a ~ i, but portions in figures only were a~f.
RESPOND:
The authors overlooked on the labelling errors while cropping and editing the respective Figure from the student’s thesis.
